# *MECP2*-Related Disorders in Males

**DOI:** 10.3390/ijms22179610

**Published:** 2021-09-04

**Authors:** Ainhoa Pascual-Alonso, Antonio F. Martínez-Monseny, Clara Xiol, Judith Armstrong

**Affiliations:** 1Fundació Per la Recerca Sant Joan de Déu, Santa Rosa 39-57, 08950 Esplugues de Llobregat, Spain; ainhoa.pascual@sjd.es (A.P.-A.); clara.xiol@sjd.es (C.X.); 2Institut de Recerca Sant Joan de Déu, Santa Rosa 39-57, 08950 Esplugues de Llobregat, Spain; antoniofederico.martinez@sjd.es; 3Clinical Genetics, Molecular and Genetic Medicine Section, Hospital Sant Joan de Déu, 08950 Esplugues de Llobregat, Spain; 4CIBER-ER (Biomedical Network Research Center for Rare Diseases), Instituto de Salud Carlos III (ISCIII), 28029 Madrid, Spain

**Keywords:** *MECP2*, Rett syndrome, *MECP2* duplication syndrome, encephalopathy, point mutation, loss-of-function, males

## Abstract

Methyl CpG binding protein 2 (*MECP2*) is located at Xq28 and is a multifunctional gene with ubiquitous expression. Loss-of-function mutations in *MECP2* are associated with Rett syndrome (RTT), which is a well-characterized disorder that affects mainly females. In boys, however, mutations in *MECP2* can generate a wide spectrum of clinical presentations that range from mild intellectual impairment to severe neonatal encephalopathy and premature death. Thus, males can be more difficult to classify and diagnose than classical RTT females. In addition, there are some variants of unknown significance in *MECP2*, which further complicate the diagnosis of these children. Conversely, the entire duplication of the *MECP2* gene is related to *MECP2* duplication syndrome (MDS). Unlike in RTT, in MDS, males are predominantly affected. Usually, the duplication is inherited from an apparently asymptomatic carrier mother. Both syndromes share some characteristics, but also differ in some aspects regarding the clinical picture and evolution. In the following review, we present a thorough description of the different types of *MECP2* variants and alterations that can be found in males, and explore several genotype–phenotype correlations, although there is still a lot to understand.

## 1. *MECP2* Gene

### 1.1. General Characteristics of MECP2

Methyl CpG binding protein 2 (*MECP2*) (OMIM *300005) encodes the protein MeCP2 and is located in the Xq28 region which can be inactivated for gene dosage compensation of the X chromosome in females [1]. *MECP2* has four exons and undergoes alternative splicing from which two well-characterized isoforms are generated—isoform e1 and isoform e2. Isoform e1 retains exons 1, 3, and 4 whereas isoform e2 retains exons 2, 3, and 4. *MECP2_e1* is conserved across vertebrates while e2 appeared later in the class Mammalia [2]. *MECP2_e1* is the most abundant isoform in the brain although the ratio between the two isoforms varies across different tissues; for example, *MECP2_e2* is more abundant in fibroblasts [2]. Even though both isoforms share the majority of their sequence and the main functional domains, they are not completely redundant. Each of them has their own properties, spatial expression, function, and interacting partners [3,4,5].

*MECP2* has several structural domains—N-terminal domain (NTD), methyl-binding domain (MBD), intervening domain (ID), transcriptional repression domain (TRD), and C-terminal domain (CTD). MBD and TRD are considered crucial functional domains. MBD enables the binding to methyl CpG dinucleotides and is where most of disease-causing mutations are located. TRD is needed for the binding and posterior recruitment of co-repressor proteins, such as NCoR, SMRT, and HDAC3, in order to repress transcription [6]. The protein has a nuclear localization signal (NLS) domain as well. MeCP2 is an unstructured protein that can adopt local secondary structures when binding to other molecules, which explains its involvement in multiple molecular interactions and thereby, functions [7,8]. Thus, *MECP2* is a multifunctional gene that acts as a transcriptional regulator (both activating and repressing) and a chromatin remodeler; it also interacts with the RNA splicing machinery and with microRNA processing machinery, among others [9]. Post-translational modifications are also implicated in regulating its activity and interactions with other proteins [10,11]. The resultant protein MeCP2 is ubiquitously expressed even though it is more abundant in the brain, especially in neuronal cells. It is noteworthy that the level of expression correlates with the maturation of neurons, indicating the importance of MeCP2 not only in neuronal development but also in neuronal maturation and maintenance [12,13].

### 1.2. One Gene, Multiple Phenotypes

Since mutations in the *MECP2* gene were first reported in 1999 in female and male patients with Rett syndrome (RTT) (OMIM #312750) [14,15], genetic alterations ranging from single nucleotide mutations to large deletions have been described and associated with RTT. As the majority of the reported cases described affected females, it was suggested that mutations in *MECP2* lead to embryonic lethality or early postnatal death in males, since no wildtype allele can be partially expressed as in females. However, sporadic reports of boys with mutations in this gene have shown otherwise [15,16,17,18,19].

Anyway, *MECP2* is not only associated with RTT, but is also related to severe neonatal encephalopathy (OMIM #300673), autism susceptibility (OMIM #300496), X-linked mental retardation syndrome (OMIM #300055), and *MECP2* duplication syndrome (MDS) (OMIM #300260). All these phenotypes, and others not so well defined, have been reported in males carrying variants in *MECP2.*

As can be inferred, *MECP2* is a dosage-sensitive gene because loss-of-function mutations lead to RTT, but whole gene duplication leads to MDS. This must be taken into consideration when looking for a treatment. Here, we would like to take a deeper look into the male cases harboring mutations and alterations in *MECP2* since much remains to be understood.

## 2. Mutations in *MECP2*

Whenever a mutation in *MECP2* is found in a patient, RTT becomes a possible diagnosis. RTT was first clinically described in 1966 by Andreas Rett in girls. In 1999, Zoghbi’s group linked *MECP2* to RTT [14,15]. Since then, groups all over the world have reported patients, reaching a few thousands of cases. In fact, *MECP2* mutations cause 97% of classic RTT cases [20,21].

Several specific *MECP2* mutation screenings have been performed in males affected by neurological disorders. In all of them, a low frequency of variants in *MECP2* was found [22,23,24,25]. RettBASE is an international curated database, which gathers the genetic variation found in individuals with RTT and related clinical disorders. To date, there have been 3924 female cases with mutations in *MECP2* and 345 male cases. As in females, in males, mutations range from single nucleotide changes to larger deletions involving up to 240 nt [26,27]. Small duplications from one to seven nucleotides have also been reported in RettBASE. The disparity of cases for each sex and the difficulties in creating the first male mouse model suggested that mutations in *MECP2* in males were lethal. Fortunately, different groups have reported new patients and, nowadays, we know that the effects of these mutations range from severe neonatal encephalopathies and premature death (as in the case of c.806delG [26]) to mild intellectual and psychomotor impairment (as in c.608C > T [19]) and that they are not always related to RTT.

### 2.1. Clinical Presentation

RTT is a very well-characterized syndrome in females with approved and revised diagnostic criteria since the early 1990s [28,29,30,31] and established severity scores [32,33,34]. However, in males, the diagnosis sometimes remains unclear. Several groups have studied these male patients and classified them according to clinical and genetic criteria (Table 1).

In 2003, Ravn et al. compared the first group of 18 male patients with pathogenic variants in *MECP2*. They classified the variants into two groups—mutations causing RTT in girls and mutations that do not affect or cause mild intellectual disability (ID) in females. This genetic classification corresponds with the phenotypes of the boys, which are divided into two groups as well—cases with severe neonatal encephalopathy and cases with non-specific mental retardation. They pointed out that, among the patients harboring RTT mutations, two kinds of patients could be found. Whenever the patient has Klinefelter syndrome (47, XXY) or is a mosaic, the boy develops an RTT phenotype. However, if the chromosomal complement is normal and no mosaicisms are found, the boy usually dies at a very early age [26].

Huppke and Gärtner classified the male patients with mutations in *MECP2* into three groups: (1) boys presenting severe encephalopathy and infantile death with *MECP2* mutations seen in RTT girls; (2) boys with RTT, who are mosaic or have a 47, XXY karyotype and harbor mutations already seen in girls with RTT; and (3) boys with less severe neurological and/or psychiatric manifestations. These mutations have not been found in RTT girls [35]. This classification was used in subsequent articles to group the cases of males with variants in *MECP2* [36,37,38]. Although the previous classifications stated that an abnormal chromosomal complement is needed to have RTT as a male, there are reported cases with a normal 46, XY karyotype and diagnosed RTT such as the one reported by Masuyama et al. [39] In those cases, due to limitations of the technique, mosaicisms cannot be excluded.

Recently, Neul et al., performed a thorough examination of males with *MECP2* mutations enrolled in the RTT Natural History Studies RTT5201 and RTT5211 [40]. Cases were divided into four groups according to their phenotype: (1) classical or atypical RTT when consensus criteria were met, (2) neonatal encephalopathy when the impairment was seen from birth, (3) progressive encephalopathy when the worsening of the clinical traits was delayed, and (4) cognitive impairment when no progressive worsening was detected during the study. They also emphasized that the clinical evolution of males meeting the diagnostic criteria for typical RTT is more severe than that observed in females. This includes more impaired initial development, ventilatory requirement, and early death. They suggested a new diagnostic category named “male RTT encephalopathy”. The new term must include all three following criteria: (1) meeting all the criteria for RTT (having a regression pattern, two of the four main criteria, and five of eleven supportive criteria), (2) having a mutation in *MECP2*, and (3) being male. This new term includes sufficient clinical features for an RTT diagnosis but acknowledges that the progression and pattern of the disease is different from that seen in females.

We encourage clinicians and researchers to start applying the classifications mentioned above depending on the kind of information possessed. Whenever the available information is clinical, Neul’s group’s classification could be used. Otherwise, if genetic information is as well known, Huppke and Gärtner’s classification could be used (see Table 1). Knowing the chromosomal complement or the mosaicism status of the patient, together with a thorough exploration considering the consensus criteria for RTT diagnosis and information about whether the mutation has already been identified in RTT girls, would improve diagnosis of these individuals and would lead to a more personalized visit. Males carrying *MECP2* variants have a very wide spectrum of clinical presentation and evolution and are difficult to classify, unlike classical RTT females.

### 2.2. Genotype–Phenotype Correlation

Improvements in the technology for molecular diagnosis have proven to be helpful. Since the discovery of *MECP2* as a causative gene of RTT, the denaturing high-performance liquid chromatography (DHPLC) technique and direct sequencing of the coding exons has been performed to confirm the clinical suspicion [18,35,41]. At the beginning, the search for alterations in the sequence was focused on exons 3 and 4 since most of the coding region is located in them. When the second isoform was found and the first mutations were reported in exon 1, the search expanded to cover all four exons [2].

The incorporation of next generation sequencing (NGS), especially of gene panels, has helped reduce the time needed for a molecular diagnosis in patients with rare diseases because of its ability to multiplex genes and patients. NGS has enabled the finding of the molecular cause in patients with either a more recognizable RTT phenotype and for whom traditional techniques were unable to detect a variation, or a more ambiguous phenotype such as X-linked intellectual disability [42,43,44]. The implementation of NGS as a diagnostic tool has found new patients with *MECP2* variations, especially males, who otherwise might never have been redirected for a *MECP2* direct sequencing test [own data]. In addition, NGS-based methods possess a high read coverage for the amplified genes which makes them a technique to take seriously into consideration for mosaicism detection rather than Sanger sequencing [45].

More and more variants of different types are being reported in *MECP2*. Some of them coincide with the mutations found in girls diagnosed with RTT. Others might seem pathological because of the effect they cause on MeCP2. But there are cases in which the relationship between the variant and the phenotype is not clear and in silico studies together with functional studies must be performed in order to assess the variant’s pathogenicity. We encourage geneticists not to dismiss a variant only because it is inherited from an apparently asymptomatic mother. XCI could explain the mild phenotype of the carrier mothers. In general, the study of different molecular levels should be taken into consideration before making conclusions about the pathogenicity of a variant.

Genotype–phenotype correlations have been difficult to determine because of the low number of reported cases. Since new cases are reported individually or in small series, usually, a comparison between the new case (or cases) and genetically similar ones is conducted in order to stablish a correlation. That approximation has led to diverse outcomes [26,46]. The lack of always having a straight correlation highlights the molecular complexity within these boys. Neul et al., 2019 found a correlation in their cohort and observed that males with early RTT mutations (before codon 271) had a higher RTT Clinical Severity Scale (CSS) score compared to the ones with later RTT mutations [40]. That could be accomplished because a large group of patients was analyzed.

## 3. Duplication of *MECP2*

Duplication of the entire *MECP2* and *IRAK1* (OMIM *300283) genes causes *MECP2* duplication syndrome (MDS). Classical MDS phenotype shows mainly in males while in females the severity ranges from anxiety, autistic features, and mild intellectual impairment to a severe phenotype similar to that reported in males (Figure 1). Although the duplication can occur *de novo*, it is usually inherited from a carrier mother with a skewed XCI who is apparently healthy.

In the late 1990s, several groups were trying to link patients with X-linked mental retardation (XLMR) to specific genetic alterations or genes. During this search, several cases with X chromosome distal duplications were reported [47]. In particular, Lubs et al., described a family of five affected boys with an Xq28 duplication inherited from carrier mothers which later were confirmed to be proper cases of MDS [48,49]. Because of that first article, MDS was originally named as Lubs X-linked mental retardation syndrome, a name that still can be found in OMIM (MIM #300260). In 2005, publications of well-characterized new cases with duplications encompassing *MECP2* led to the establishment of the MDS as we know it nowadays [47,50,51].

More publications with further MDS cases followed those first articles and scientists and clinicians started to wonder about the frequency of the syndrome. Shao et al., Ramocki et al., and de Brouwer et al., analyzed the results of 5380 and 4683 male cases and 600 family members, respectively; the frequency of MDS was 0.30%, 0.41%, and 0.5% [52,53,54]. Shao et al., pointed out that duplication of Xq28, including *MECP2*, was the most common duplication they found in their cohort. Whereas the screening of males with an X-linked inheritance and idiopathic XLMR without expansion of *FMR1* and normal karyotypes set the frequency of MDS at 1%, the screening of males with ID and severe, mostly progressive, neurological symptoms raised the frequency to 2.2% [55]. In a similar cohort of multiple congenital anomalies with ID, Honda et al., set the frequency of MDS at 2.6%, and among families with ID, 2.3% [56]. The frequency of MDS in small male cohorts with specific inclusion criteria ranges from 1.6% to 17.6% [53,57,58].

Even though MDS is a rare syndrome and most of the articles describe sporadic or small familiar cases rather than large cohorts, to date, there are more than 600 cases reported worldwide [47,48,49,50,51,55,56,57,58,59,60,61,62,63,64,65,66,67,68,69,70,71,72,73,74,75,76,77,78,79,80,81,82,83,84,85,86,87,88,89,90,91,92,93,94,95,96,97,98,99,100,101,102,103,104,105,106,107,108,109,110,111,112,113,114,115,116,117,118,119,120,121,122,123,124,125,126,127,128,129,130,131].

### 3.1. Clinical Presentations

MDS is characterized by a wide variety of symptoms. The core phenotype of MDS patients includes hypotonia, developmental delay (DD), mostly moderate-severe intellectual disability, autistic features, epilepsy, progressive lower extremity spasticity, poor speech development, recurrent infections, and gastrointestinal problems. Descriptions of individual cases and, specially, of large cohorts have helped us better understand the main and constant symptoms of MDS and expand its phenotype [53,102,109,113,116,119,121,123,129,130,132]. Even so, there are no publications focused on knowing the natural history of this disorder.

#### 3.1.1. Neurological Aspects

Hypotonia, which is only absent in few children, is associated with delay in attaining developmental milestones including sitting and crawling. Some males achieve ambulation without support, but it is usually an ataxic gait that generates a lumbar hyperlordosis as a compensation [132]. Unfortunately, with age, many MDS patients manifest a motor regression that will lead to progressive loss of ambulation in some of them [116,121,132].

Many individuals with MDS meet the formal criteria for a diagnosis of autism spectrum disorder due to poor expressive language skills, abnormal social affect, and restricted/repetitive behaviors. Mood disorders such as anxiety sometimes occur. Two studies comparing a group of 9 and 10 boys with MDS and a matched group of boys with idiopathic autism spectrum disorder (ASD) revealed similarities between them [70,133]. Another study, however, highlighted that a considerable proportion of their MDS cohort lacked social impairment since they interacted during the consultations and that they frequently smiled meaningfully [116]. Moreover, most affected children do not develop speech and the ones who at some point speak a few words usually lose that ability.

Hand stereotypies and bruxism are traits shared between RTT and MDS [53,109,116,130]. However, while hand stereotypies in RTT replace purposeful hand use, they do not in MDS and, when present, they appear later than in RTT [8,116].

It is believed that refractory epilepsy and recurrent infections trigger the regression in these patients. Seizure types that have been reported in individuals with MDS include head/neck and trunk-drop attacks, absence seizures, myoclonic seizures, and generalized or secondarily generalized tonic-clonic or simply tonic seizures (once known as grand mal seizures). Up to half of individuals develop recurrent seizures in childhood or adolescence. Seizures may start at a very variable age and the prevalence might be underestimated, since, in some patients, the onset occurs during the second decade of life [132]. Unfortunately, more than half will possess drug-resistant epilepsy [116]. A decreased pain sensitivity has been found in these children as well [116,123].

#### 3.1.2. Immunological Aspects

Together with epilepsy and neurological deterioration, recurrent infections, mainly in the respiratory tract, are the main cause of concern and are a major contributing factor to reduced life-expectancy. Affected individuals may develop recurrent pneumonia that is sometimes severe, requiring mechanical ventilation. This is responsible for the many hospitalizations and the premature death of many MDS patients. Middle ear infections (otitis media) and sinusitis are also common. Additional infections have been reported including meningitis or urinary tract infections. Bauer et al., recommend vaccination against pneumococci and evaluation of post-vaccination titers [98]. MDS patients show irregularities in their antibody levels, sometimes even after shorts periods of time post-vaccination; in those cases, extra boosters are required [98,115,132]. Nevertheless, there are patients who have shown a reduction in the frequency of infections with age [67].

#### 3.1.3. Gastrointestinal Aspects

Feeding difficulties due to hypotonia may be evident shortly after the child is born. Frequent gastrointestinal problems are gastro-esophageal reflux, swallowing dysfunction, and excessive drooling, which may contribute to their recurrent respiratory infections. Affected children will often fail to gain weight or will grow at the expected rate for age and gender but may be at a risk of aspiration; however, some infants experience no recognized problems in the neonatal period and concern is not raised until other developmental milestones are missed. The majority of patients suffer from severe constipation [109,130]. Bladder dysfunction has also been seen. Fortunately, Miguet et al., observed in their cohort that, with age, these difficulties often disappear [116].

#### 3.1.4. Dysmorphological Aspects

MDS patients have some dysmorphic features as well. Among them are brachycephaly, deep-set eyes, strabismus, midface hypoplasia, a small open mouth, thick lower lip, large prominent ears, prominent nasal bridge, pointed nose, prominent chin, thick and dense hair, teeth anomalies, and tapered fingers [116,130]. Early in life, head size appears to be normal, but as the boys get older, some might develop microcephaly. However, macrocephaly has also been observed, and overall, these are not consistent features of the syndrome. Some boys have hypoplastic genitalia, cryptorchidism, and hypospadias. Some degree of growth retardation has been described as well. Lim et al., found that 21.7% of their studied males suffer from scoliosis, a trait that they estimated half of the patients will develop by the age of 22 [109].

Due to the potential facial gestalt of MDS male patients, the Face2Gene platform could be trained to identify MDS patients. Through automated image analysis, Face2Gene (FDNA., Boston, MA, USA) has been widely used as the tool based on pattern recognition of frontal photographs in the field of syndromic rare diseases (https://face2gene.com (accessed on 14 June 2021)) [134]. This work consisted of training of the algorithm within the tool from frontal facial photographs of MDS males and females and observing differences between them.

#### 3.1.5. Evolution of the Syndrome with Age

As mentioned, some clinical features, such as epilepsy, respiratory tract infections, or constipation, appear or worsen with age. As a result, Peters et al., reported that their older participants have more severe clinical symptoms, specifically regarding motor dysfunction (e.g., dystonia, scoliosis, and/or rigidity) and functional skills (e.g., motor skills, communication skills, chewing, and swallowing) [129]. A longitudinal Japanese study found a similar outcome by comparing the clinical traits of MDS boys at their first medical visit and some years later [130].

### 3.2. Characteristics of the Duplication

Duplications can be located at different genomic positions. They have been reported in tandem in the same Xq28 region, translocated to the Xp arm, or even outside the X chromosome [47,87,92,123]. Cases of *MECP2* triplications have also been described in males [57,68,82,84,93,104].

Duplications can be inherited from mothers to their children and the location and gene content is mainly maintained. However, detailed studies of the duplications between different families have revealed that MDS duplications are non-recurrent, as each rearrangement is of a different size and has different breakpoints [62,68,72,82,97,123]. Moreover, Yi et al., found that the transmission of the duplication is not always stable and can increase or decrease in size when passed from mothers to children [105]. The Xq28 region is an unstable region because of the high GC content and the elevated number of repetitive elements as Alu or low copy repeats (LCR). These factors are involved in the breakpoints that lead not only to DNA duplications around *MECP2* but also to deletions and inversions within it [62,135]. Nevertheless, what all the duplications of MDS have in common is that they always contain the genes *MECP2* and *IRAK1*, which comprise the minimal duplicated region. Reported duplications range from 0.079 Mb to 15.8 Mb [87,92].

In order to detect duplications (or triplications) in the Xq28 region and to establish their length, most laboratories use the microarray-based comparative genomic hybridization (array-CGH) as it is broadly used to study patients with developmental delay [132]. Real-time quantitative PCR and MLPA techniques can also be used to diagnose or to confirm the result of an array-CGH. The location of the extra copy can be determined by performing a FISH study.

### 3.3. Genotype–Phenotype Correlation

Establishing a genotype–phenotype correlation has been a challenge in MDS because of the lack of large cohorts and long-term studies on them. Several traits have been taken into consideration.

Males with *MECP2* triplication seem to be more severely affected, which points to gene dosage as a contributor to the severity in MDS [57,82,84]. A similar result has been seen in girls [81].

The location of the duplications has been studied as well. It was thought that girls with translocations of the duplications to autosomal chromosomes were more severely affected than girls with interstitial duplications, since the former escape XCI [74,106]. However, females with duplications in tandem and a severe phenotype have also been reported [86]. In our personal experience with a Spanish cohort, a boy with a duplication in the Y chromosome was the most severely affected individual and died before one year of age. Nevertheless, we know another boy with an even larger duplication in the Y chromosome who is still alive [123].

The size of the duplication has not succeeded in grouping patients according to their clinical severity [55,57,68,70,87,92,94,105,116,123]. Recently, Peters et al., studied a cohort of 48 MDS individuals and applied the CSS used in RTT [122]. They saw a correlation between CSS and severity. It should be noted, though, that a specific scale would have been more adequate since the phenotype of MDS and RTT, especially in males, has little overlap. A bespoke scale that incorporates the actual symptoms and complications of MDS patients is required to help describe the natural history of the disorder, establishing genotype–phenotype correlations and monitoring the evolution and the response to future potential treatments within clinical trials.

The role of *MECP2* as the main causative gene of the syndrome is clear. Furthermore, it has been hypothesized that the gene content of each duplication could contribute in different degrees to the clinical phenotype. *IRAK1*, which is part of the minimal duplicated region, encodes for the interleukin-1 receptor-associated kinase 1, which participates in the TLR/IL1R signaling pathway and its overexpression could be related to recurrent infections [98]. The *FLNA* (OMIM *300017) gene could contribute to the intestinal pseudo-obstruction problems of these patients [66]. Loss of function of *L1CAM* (OMIM *308840) is associated with hypoplasia of the corpus callosum, which is a common finding in MDS brain imaging [102,136]. However, larger cohorts should be studied in order to find a genotype–phenotype correlation for this trait. *RAB39B* (OMIM *300774) has been highlighted as well, since duplications of it are related with ID. In the study conducted by Peters et al., they found that MDS patients with duplications harboring *RAB39B* have higher CSS [122].

There are cases of relatives sharing the same duplication and presenting a variable phenotype [97,105,113,117,123]. Interestingly, somatic mutations have recently helped explain the phenotypic differences between monozygotic twins [117]. This event adds more complexity to the current challenge of establishing a genotype–phenotype correlation in small cohorts of rare diseases such as those gathered in this review. The possibility that the clinical phenotype of these patients varies due to the disruption of specific genes or their regulatory regions cannot be ruled out. In any case, further studies are still needed in order to obtain robust results about other genes that contribute to the phenotype.

## 4. Modeling RTT and MDS for Future Therapies

We would not like to end this review without briefly mentioning the advantages of in vitro and preclinical models. The difficulty in accessing target tissue samples from children affected by neurodevelopmental disorders has encourage researchers to create specific animal and cellular models to gain knowledge of rare diseases as RTT and MDS. In RTT, the most frequently used animal model has been the male *Mecp2*-null mouse (*Mecp2^−/y^*) which manifests the early severe phenotype seen in humans [137]. Despite being the major source of findings related to mechanisms and pathways in RTT, the translatability of mouse models towards humans is not clear, especially regarding RTT females. Several mouse models for MDS have also been created [138,139]. Alternative models, such as primary cultures of patients’ peripheral tissues, human embryonic stem cells (hESCs), or human-induced pluripotent stem cells (hiPSCs) reprogrammed from patients’ somatic cells, have proven to be very useful [103,140]. Tang and colleagues found that in hiPSCs of a male RTT patient, elevating KCC2 levels could ameliorate the functional deficits caused by the absence of MeCP2, and showed that IGF1 treatment works in the mentioned tissue [141,142]. Kim et al., also found that in RTT hiPSC knockdown of LIN28 expression partially reversed the synaptic deficits [143]. Recently, 3D aggregates from hiPSCs have been developed in an attempt to mimic the complex architecture and functions of organs such as the brain [144]. Moreover, region-specific brain organoids have been generated. The created organoids have also proven to be mutation-dependent and different initial phenotypic alterations have been found in organoids with different backgrounds [145]. All these in vitro human-derived models seem truly promising, not only because of the molecular and genetic insight they are generating, but also because several drugs are being tested and these could, ultimately, undergo clinical trials.

Even though most of the clinical trials for RTT have female participants, according to the register of the U.S. National Library of Medicine a few clinical trials have incorporated male patients. Such is the case of NCT00593957, NTC01520363 with dextromethorphan, NCT02790034 with sarizotan, and NCT00299312, in which a phase of genetic and physical characterization of RTT patients has been done. On the other hand, there are no clinical trials registered yet for MDS, but some promising results have been obtained in the previous in vitro and animal models. Recently, Ash et al., found that the hyperactivity seen in ERK the pathway in MDS could, similar to other autism-associated disorders, be reversed with ERK-specific pharmacologic inhibitors [146,147]. Moreover, antisense oligonucleotide (ASO) therapies are showing promising results in mice, especially because of their ability to reduce MeCP2 expression in a dose-dependent manner [139]. These results show that CNS administration of *MECP2*-ASO is well tolerated and beneficial in a mouse model. Although these first studies do not include the *IRAK1* gene, they provide a translatable approach that could be feasible for treating MDS. The CRISPR-Cas system has been tested in animal models and human primary fibroblasts and has successfully corrected the duplication of *MECP2* including *IRAK1* [140]. However, there is not enough evidence so far to suggest possible approaches to therapy targeted the pathophysiology underlying these two diseases. Further work could bring deep brain stimulation, ASO, and gene therapy into the clinic within the coming decades [8].

## 5. Conclusions

RTT is a very well-defined syndrome in females and, when the required main and supportive criteria are met, the diagnosis becomes clear and accurate. In males, however, the clinical manifestations generated from variants and mutations in *MECP2* are so different that a diagnosis is not always reached. Classification of these male patients into the mentioned groups should help clinicians and geneticists to better understand the phenotypes that arise from alterations in *MECP2* and to establish the molecular basis for the genotype–phenotype correlation. In addition, technology is rapidly evolving and worldwide databases with detailed information are helping us understand and interpret the new or rare variants that are found in *MECP2.* It was proven that *MECP2* in males is also related to neurodevelopmental phenotypes, thus, we encourage geneticists not to exclude a variant in this gene without performing further studies, both molecular and functional.

Contrary to mutations in *MECP2*, duplication of the entire gene in males is associated with well-defined MDS. Even though there are several clinical and molecular aspects of it that are still unknown, further studies with large cohorts, such as the recent ones we have discussed here, will be promising. The platform Face2Gene will surely help decrease the time for an accurate diagnosis of MDS. Moreover, the implementation of a specific MDS scale will be of great value, not only to describe the status of each patient and to establish a genotype–phenotype correlation, but also to monitor the response and evolution of future clinical trials.

## Figures and Tables

**Figure 1 ijms-22-09610-f001:**
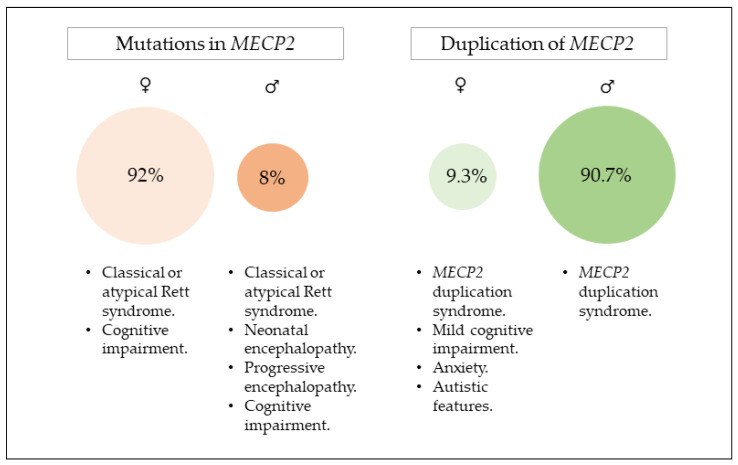
Frequency of each sex among patients harbouring mutations in *MECP2* and those with a duplication of the entire *MECP2*. Sources of information are RettBASE for mutations in *MECP2* and 82 articles from PubMed containing the term “*MECP2* duplication” for the patients with the duplication.

**Table 1 ijms-22-09610-t001:** Different methods of classifying male cases with mutations in *MECP2* depending on clinical and/or genetic criteria.

Classification Groups	Reference
1. **Boys with severe neonatal encephalopathy.** When having normal chromosomal complement (46, XY) they die at an early age. When being mosaic or having Klinefelter syndrome (47, XXY) they develop classic RTT. The same mutations cause RTT in girls. 2. **Boys with non-specific mental retardation.** They have normal chromosomal complement. The same mutations do not affect girls or cause mild ID.	Ravn et al., 2003
1. **Boys with RTT.** RTT consensus criteria are fulfilled. They are mosaic or have Klinefelter syndrome (47, XXY). The same mutations cause RTT in girls. 2. **Boys with severe neonatal encephalopathy and early death.** They have a normal chromosomal complement (46, XY). The same mutations cause RTT in girls. 3. **Boys with less severe neurological and/or psychiatric manifestations.** They have normal chromosomal complement. The same mutations do not affect girls, cause mild ID, or have not been reported in female patients.	Huppke and Gärtner 2005, Moretti et al., 2006, Villard 2007, Neul et al., 2012
1. **Boys with Classical or atypical RTT.** Consensus criteria for each category are fulfilled. When classical RTT is diagnosed the term “male RTT encephalopathy” is suggested. 2. **Neonatal encephalopathy.** Impairment of the clinical traits is noted from birth. 3. **Progressive encephalopathy.** Impairment of the clinical traits appears over the years. 4. **Cognitive impairment.** No progressive worsening is detected.	Neul et al., 2019

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
