# Peer review of "MECP2-Related Disorders in Males"

_ijms, 2021, doi:10.3390/ijms22179610_

Round 1

Reviewer 1 Report

The manuscript from Pascual-Alonso et al. gives an overview over the genetic and clinical aspects of MECP2-related disorders in males. This is a very interesting and important topic since while in females the consequences of mutations in MECP2 are well described, in males these are still largely undescribed resulting often in unclear diagnosis. Thus, while the review addresses an important topic, it presents a few weak points that need to be improved.

  • The review is in parts written in a repetitive manner. This together with the high density of information makes it difficult to fully appreciate the content.
  • Moreover, part of the reviews are written in an almost too colloquial manner (e.g. “a lot of”…”; “…having a good appetite”; “our clinical geneticists…”)
  • I am not sure whether the title “Point mutations in MECP2” is precise enough. As I understand this part reports not only on point mutations, but on all loss-of-function mutations. However, this is not very clear in the text
  • There are a couple of references to not yet published own data (manuscript in preparation). I am not sure what is the policy of the journal here, but I find difficult to include in a review information that has not yet undergo peer review. This, in my opinion adds to the anecdotical impression given by parts of this review.
  • The authors describe a few recently published attempts at classifying male patients according to clinical and genetic criteria and they stress the importance to start using these classifications. However, the classifications described are slightly different it would be interesting to include a more critical analysis of which criteria / classifications are recommended to use

Reviewer 2 Report

In this review, the authors did a systematically description of the different types of MECP2 variants and alterations observed in male patients, by describing several genotype-phenotype correlations observed in MECP2 point mutations and in MECP2 duplication. The topic falls within the scope of journal. I would like to suggest the authors to modify few issues and to add some more details/info in order to increase the interest and complexity of the topic, before considering appropriate for publication. Please see the comments.

Comments:

  1. I would like to suggest the authors to add some examples of MECP2 specific mutations considered to be responsible for more severe or for mild Rett male phenotype, as mentioned in the following sentence, (starting at line 89). “Fortunately, different groups have reported new patients (…) severe neonatal encephalopathies and premature death to mild intellectual and psychomotor impairment”.
  2. On the topic - clinical presentation, it would be interesting if the authors sum up the information in a table (related with the several classifications /groups (with the respective ref.) according to the Rett male patient’s phenotype).
  3. A simple table could help in summing up the distinct and similar features regarding the MECP2 point mutation variants and the MECP2 duplication phenomenon.
  4. The topic: 3.1.5. Other aspects - is vague and does not add much as topic. The content may be included within other topic.
  5. The topic: 3.1.6. Evolution - may have a more complete title, and the included information regarding the evaluative phenotypic scenario of Rett male patients must be better developed.
  6. In the present review, the authors described different types of MECP2 variants, by describing genotype-phenotype correlations in patients. However, the review could include some more data/topics that could greatly increase the content, complexity and interest of the subject related with MECP2 alterations in male patients. a) It could be interesting if the authors also include a topic (before the conclusion) related with in vitro/preclinical studies in both scenarios: Rett male point mutations models and on MECP2duplication syndrome models. I would like to suggest a brief discussion/review related with the more recent breakthroughs using both cellular models and animal models in the study of Rett male phenotypes. I would like to suggest some papers related to the topic, which may help and may be included. https://doi.org/10.7554/eLife.02676.002, doi: 3389/fcell.2020.610427, https://doi.org/10.1073/pnas.1524013113, https://doi.org/10.3390/ijms22073751,
    https://doi.org/10.1371/journal.pone.0212553, doi: 10.1523/ENEURO.0056-21.2021, doi: 10.1523/ENEURO.0282-20.2020, doi: 10.1038/s41583-018-0006-3. b) It could be interesting if the authors also include a topic, before the conclusion, related with ongoing/completed clinical trials focusing on studies with male participants, describing the more relevant objectives, details and pointing out possible outcomes observed. Some examples: NCT00593957, NCT02061137, NCT01520363, NCT00299312, NCT02790034.

Minor comments:

  1. On the topic “General characteristics of MECP2”, the first sentence should have a reference. On line 65, “However, sporadic reports of boys with mutations in this gene have shown otherwise.” Some references of the stated “reports” must be added.
  1. The XXY karyotype, is also known as the Klinefelter syndrome. The authors should complete this information.

Reviewer 3 Report

This paper is well structured and interesting and significant for clinical knowledge. However, it would be useful to introduce a small section devoted to current and possible medical treatments for this syndrome. 

Author Response

Reviewer: 3
This paper is well structured and interesting and significant for clinical knowledge. However, it
would be useful to introduce a small section devoted to current and possible medical
treatments for this syndrome.
We agree with your comment and we added a new paragraph about possible medical
treatments for this syndrome.
We think that the revised version addresses all the major concerns of the referees. We
thank you for your offer to review this paper, and hope that it will be acceptable for
publication in International Journal of Molecular Sciences.

Round 2

Reviewer 2 Report

I would like to thank the authors, which successfully improved the manuscript.

I suggest that the manuscript is appropriate for being published.

Congratulations!